# Symptoms of Discomfort and Problems Associated with Mode of Delivery During the Puerperium: An Observational Study

**DOI:** 10.3390/ijerph16224564

**Published:** 2019-11-18

**Authors:** Juan Miguel Martínez-Galiano, Miguel Delgado-Rodríguez, Julián Rodríguez-Almagro, Antonio Hernández-Martínez

**Affiliations:** 1Department of Nursing, University of Jaen, 23071 Jaen, Spain; jgaliano@ujaen.es; 2Consortium for Biomedical Research in Epidemiology and Public Health (CIBERESP), 28029 Madrid, Spain; mdelgado@ujaen.es; 3Department of Health Sciences, University of Jaen, 23071 Jaen, Spain; 4Department of Nursing, Faculty of Nursing of Ciudad Real, University of Castilla-La Mancha, 13071 Ciudad Real, Spain; antomatron@gmail.com; 5Department of Obstetrics & Gynecology, Mancha-Centro Hospital, 13600 Alcázar de San Juan Mancha, Spain

**Keywords:** mode of delivery, puerperium, health problems, discomfort, maternal problems, self-perceived health, birthing plan, policy

## Abstract

Despite abundant literature on antenatal and delivery care received by pregnant women, there is a wide knowledge gap on the prevalence of symptoms of discomfort or problems during the postpartum period and their relationship with the mode of delivery. This cross-sectional study, carried out with 3324 participants in Spain in 2017, aimed to investigate the association between the mode of delivery and self-reported postpartum symptoms of discomfort and maternal problems during the puerperium. An ad hoc online questionnaire was used to collect data on socio-demographic and obstetric variables, symptoms of discomfort, and maternal problems during the puerperium. The crude odds ratios (OR) and adjusted OR (aOR) and their 95% confidence intervals (95%CI) were calculated using binary logistic regression. In total, 3324 women participated. Compared to a normal vaginal delivery, having a cesarean section was associated with increased odds of an infected surgical wound (aOR: 11.62, 95%CI: 6.77–19.95), feeling sad (aOR: 1.31, 23 95%CI: 1.03–1.68), and symptoms of post-traumatic stress (aOR: 4.64, 95%CI: 2.94–7.32). Instrumental delivery vs. normal vaginal delivery was a risk factor for constipation (aOR: 1.35 95%CI: 25 1.10–1.66), hemorrhoids (aOR: 1.28, 95%CI: 1.04–1.57), urinary incontinence (aOR: 1.30, 95%CI: 26 1.05–1.61), and fecal incontinence (aOR: 1.94, 95%CI: 1.29–2.92) during the puerperium. Women who gave delivery via cesarean section or instrumental delivery had higher incidences of infection and psychological alterations than those who had a normal vaginal delivery. Identifying women at risk of giving birth by cesarean section and informing them about subsequent symptoms of discomfort and maternal problems during the puerperium must be included in pregnancy health program policies and protocols to allow women to make informed decisions regarding their birthing plan.

## 1. Introduction

During pregnancy, childbirth, and the puerperium, events may arise that affect a women’s health in physical, psychological, and social terms, not only while pregnant and during childbirth, but also after giving birth, and these may develop into chronic illnesses [1,2,3,4,5,6,7].

Some factors related to pregnancy and birth, as well as certain clinical practices performed during childbirth, are associated with greater maternal morbidity [8,9,10,11,12]. One of the studied factors is the mode of delivery by which women give birth. The mode of delivery has been associated with various maternal complications during the puerperium and also in the long term [13,14,15,16,17,18,19]. One study conducted in China with 7046 women found that a cesarean section vs. instrumental delivery with forceps was associated with increased incidence of maternal infection during the puerperium, but less postpartum hemorrhage [13]. In England, a study by the U.K. National Maternity Survey involved 5332 women. In this study, an instrumental delivery with forceps or non-programmed cesarean section was associated with worse postpartum psychological health compared to those who had normal vaginal deliveries or programmed cesarean sections [15]. Maternal postpartum sexual relations were studied as part of a systematic review involving a meta-analysis of 10 studies that included a total population of 2851 women, and no association was found between mode of delivery and maternal postpartum sexual relations [14]. However, a cross-sectional study with 238 women in Lebanon reported that the women who had given birth via cesarean section reported more painful sexual intercourse during the postpartum period [17]. A study by Sword et al. in Norway suggested that the mode of delivery had no significant impact on the development of postpartum depression in the main-effects model. However, it may interact with place of birth and other unmeasured factors to create a risk for depression [19].

Sometimes women request giving birth by a specific mode of delivery. Some prefer giving birth as naturally as possible with no medication, as set out by the World Health Organization (WHO) [20]. However, more women request giving birth via cesarean section, with no clinical indication to justify this decision other than not wishing to undergo vaginal delivery [21,22,23,24,25]. Among women who had a cesarean section, between 1–48% in the United States and 18% in the United Kingdom were at the request of the mother herself [26]. In Spain, there is also a considerable number of cesarean sections performed without clinical indication, instead happening at the mother’s request [27]. In this sense, it is possible that many professionals and women are unaware of the risks, the symptoms of discomfort, and the problems that one type of delivery or another can entail.

The effect of the mode of delivery on different postpartum health indicators has been studied, but with inconsistent findings; our study intends to explore and verify these findings. Moreover, in Spain there are very few studies that address this issue [13,14,15,16,17,18]. Thus, research into this matter is needed. Overall, a tendency exists for women to request the mode of delivery they wish, and more cesarean sections are being requested with no clinical indications to do so. Knowing the effects of the mode of delivery on a women’s postpartum health may make them aware of the consequences of the mode of delivery they choose and make them able to decide responsibly. For this reason, our objective was to establish the association between different modes of delivery (normal vaginal, instrumental, and cesarean section) and the issues women may experience afterwards.

## 2. Material and Methods

### 2.1. Design and Selection of Study Subjects

A cross-sectional study was conducted with women who gave birth in Spain in 2017 (both in public health system centers and in private centers). Antepartum stillbirths and births by women under 18 years of age were excluded. The participants were women who gave birth in public or private hospitals throughout the Spanish territory; no woman who gave birth at home participated.

In order to estimate sample size, the maximum modeling criterion was considered, which involves 10 events (complications) for each independent variable in the multivariate analysis. By taking fecal incontinence as the least frequent complication (4%), a minimum of 2500 women were needed to include a minimum of 10 independent variables [28].

### 2.2. Information Sources

To collect information, an online questionnaire developed by the authors was used, which included 35 items (3 open, 32 closed). It was handed out to the participating women 6 weeks after giving birth. Information was collected on clinical and socio-demographic characteristics, obstetric outcomes, on newborn variables, and on any women’s problems, troubles, and requirements during the postpartum period. This questionnaire was previously piloted. The Spanish Federation of Midwives Associations (FAME) and its member associations disseminated the questionnaire to midwives throughout the Spanish territory, who recruited and encouraged women to participate. After recruiting the women in hospitals and getting them to sign informed consent to participate, they were instructed on how to fill in the questionnaire in their own time. A telephone number and a chat service were set up to help them if any doubts arose when completing the questionnaire. The variables shown below were collected.

The main independent variable was the mode of delivery the women underwent (normal vaginal, instrumental, or cesarean section).

The main outcome variables were symptoms of discomfort and self-reported maternal problems, including constipation, hemorrhoids, infected wounds (requiring professional treatment after being discharged from hospital or antibiotics treatment), perineal pain, headache, chest pain, back pain, a stinging feeling while urinating, fecal incontinence (unable to control feces), urinary incontinence (involuntary loss of urine), feeling tired, sad, or anxious (nervousness or uneasiness), depression (depressing moods), symptoms of post-traumatic stress disorder (having nightmares about birth or negatively reliving the experience of giving birth constantly), problems with having sexual relations, couples having problems after giving birth, type of newborn feeding, and problems related to lactation.

We used a variety of clinical and socio-demographic variables as controls in each of the bivariate analyses.

### 2.3. Statistical Analysis

Firstly, a descriptive analysis was run using absolute and relative frequencies. This was followed by a bivariate analysis between mode of delivery and the main symptoms of discomfort and problems. The crude odds ratio (OR) and its 95% confidence interval (95%CI) were estimated. Finally, a multivariate analysis was carried out using binary logistic regression with the potential confounding variables for each analysis.

Significance was set at *p* ˂ 0.05, and the SPSS v24.0 statistics package (SPSS Inc., Chicago, IL, USA) was used for all of the analyses.

### 2.4. Ethical Approval

This study received approval from the Ethics Committee in Clinical Research (CEIC) of the La Mancha Centro Hospital under ethical code 69-C 07/2017. Before women filled in the study questionnaire, they received information about our study and its aims. Participants had to tick a box if they consented to participation (i.e., they signed an ad hoc digital informed consent form).

## 3. Results

In the present study, 3324 women participated. of whom 51.90% (1725) were primiparous and 50.1% (1664) of them gave birth under the age of 35 years. Most of the women in our sample were Spanish (96.6%), as shown in Table 1, which describes the study population’s characteristics. This table also shows that most women (86.1%) had a low-risk pregnancy; that is, they had no health problems during pregnancy.

Table 2 shows a higher risk for twin pregnancy in association with cesarean sections (OR: 8.60, 95%CI: 5.45–13.56) and instrumental deliveries (OR: 2.34, 95%CI: 1.27–4.32) vs. normal vaginal deliveries. With regards to induction of labor, a positive association was found for cesarean section (OR = 2.36, 95%CI: 1.99–2.80) and instrumental delivery (OR: 1.63, 95%CI: 1.34–1.97) vs. normal vaginal delivery.

### 3.1. Symptoms of Discomfort and Maternal Problems Associated with Mode of Delivery

#### 3.1.1. Instrumental vs. Normal Vaginal Delivery.

Instrumental delivery vs. normal vaginal delivery was found to be a risk factor for suffering the following ailments during the puerperium: constipation (aOR: 1.35, 95%CI: 1.10–1.66), hemorrhoids (aOR: 1.28, 95%CI: 1.04–1.57), infected wounds (aOR: 2.45, 95%CI:1.61–3.71), urinary incontinence (aOR: 1.30, 95%CI: 1.05–1.61), fecal incontinence (aOR: 1.94, 95%CI: 1.29–2.92), and symptoms of post-traumatic stress disorder (aOR: 2.74, 95%CI: 1.90–3.97), as shown in Table 3.

#### 3.1.2. Cesarean Section vs. Normal Vaginal Delivery

Compared with a normal vaginal delivery, a cesarean section showed a positive association with an infected surgical wound (aOR: 11.62, 95%CI: 6.77–19.95), headache (aOR: 1,76 95%CI:1.43–2.18), feeling sad (aOR: 1.31, 95%CI: 1.03–1.68), anxiety symptoms (aOR: 1.36, 95%CI: 1.05–1.76), symptoms of depression (aOR: 1.67, 95%CI: 1.21–2.34), symptoms of post-traumatic stress (aOR: 4.64, 95%CI: 2.94–7.32), and formula feeding (aOR: 2.16, 95%CI: 1.74–2.69) during the puerperium period (Table 3). A negative association was found between cesarean section and different problems during the puerperium: urinary incontinence (aOR: 0.32, 95%CI: 0.25–0.42), a stinging feeling while urinating (aOR: 0.62, 95%CI: 0.47–0.82) and having hemorrhoids (aOR: 0.43, 95%CI: 0.35–0.52), among others (Table 3).

## 4. Discussion

### Main Findings

The present study assessed the association between the mode of delivery and various maternal parameters in the puerperium. According to our results, compared with normal vaginal delivery, having a cesarean section was associated with wound infection, headache, back pain, maternal feelings of sadness, anxiety and depression, as well as tiredness and symptoms of post-traumatic stress disorder during the puerperium period. A negative association was also observed with having perineal pain and hemorrhoids. Compared with normal delivery, our results also detected a positive association between having an instrumental delivery and having hemorrhoids, constipation, wound infection, perineal pain, and headaches during the postpartum period, as well as symptoms of post-traumatic stress and problems with sexual relations. Compared with normal delivery, we also identified instrumental delivery as a risk factor for urinary or fecal incontinence.

In terms of possible limitations of the present study, if a selection bias was associated with non-response, it did not affect our results. The majority of the women responded positively to participation and only 29 refused. There was nothing to suggest that the non-responding women would have acted differently from those who did. It is unlikely that an information bias exists; the collected data and the answers were presented in such a way that anyone with any level of education could understand them, as they were presented simply. We cannot exclude an memory bias, however, information was collected over a short period, and therefore if this bias had any influence on the results, we believe it would have been weak. The participating women remembered details about their birth process, and most paid attention and evaluated the whole process very well. The women who agreed to participate completely filled in all the questions. We cannot rule out residual confounding variables, although any influence on the results would have been minimal, as when fitting the regression model each outcome variable was adjusted individually and specifically for all possible variables that could have influenced them. These confounding variables were based on those found in a literature review and the clinical experience of the researchers. For example, one of the variables for which it was adjusted was parity. Parity has been associated with problems and discomfort in the postpartum period [29].

This study has some noteworthy points, including its large sample size, the inclusion of women who went to both private and public hospitals, and those who came from different geographical areas. Thus, it includes a wide range of societal groups and social demographics.

According to our results, multiple pregnancy, induction of labor, having health problems during pregnancy, and using epidural analgesia during delivery were identified as risk factors for not giving birth vaginally. These factors are in line with results of previous studies and have been well-studied in the scientific literature [29,30,31,32,33].

A cesarean section was found to have a protective effect for urinary incontinence, similar to the results of a systematic review meta-analyses by Rørtveit and Hannestad [34], who identified a cesarean section as a protective factor. Chang et al. also found an association between normal vaginal delivery and urinary incontinence [35]. Moreover, we associated instrumental delivery with urinary incontinence, as did Bozkurt et al. [36]. Bozkurt et al. [36], similar to Memon and Handa [37], observed no association between mode of delivery and fecal incontinence. In contrast, our results indicated an association between instrumental delivery and fecal incontinence.

Those women who underwent instrumental delivery or cesarean section reported having significantly more headaches during the postpartum period than those who had a normal vaginal delivery. These results coincide with those published by Nikpour et al. in their study in Iran involving 300 women [38].

Conversely, the study by Blomquist et al. involving 1115 women (577 cesarean sections and 538 normal deliveries) concluded that there were no differences in the perineal pain that women experienced according to their mode of delivery [39]. Our results showed an association with instrumental delivery and suffering more perineal pain compared with those who had a non-instrumental vaginal delivery and found a negative association between cesarean section and perineal pain compared to those who had a normal delivery. These findings are similar to those reported by Thompson et al., who found that instrumental deliveries were associated with more perineal pain [40].

Those women who had cesarean section or an instrumental delivery suffered more postpartum wound infections, which agrees with the findings of other authors [13]. However, a systematic review with meta-analyses identified a positive association only for cesarean sections and postpartum infection [41].

We also found an association between instrumental delivery or cesarean section and women having sexual problems during the postpartum period. These results are in line with those indicated by other authors [17], but contrast with those identified by others [14,39].

The women who had an instrumental delivery or cesarean section reported more feelings of anxiety during the puerperium period, while women who had a cesarean section reported feeling tired, in line with the results of Woolhouse et al. [42]. They also reported more feelings of sadness and having postpartum depression, which agrees with the findings of Rowlands et al. [15].

Experiencing a dystocic delivery, either by cesarean section or instrumental delivery, meant feeling more symptoms of post-traumatic stress compared to women who had a normal delivery, which is in agreement with the findings of Srkalović et al. [43]. However, a longitudinal study involving 240 women in Iran by Mahmoodi et al. [44], as well as the findings by Rowlands et al., showed an association between instrumental delivery and post-traumatic stress [15].

We found a negative association for cesarean section and urinary problems, which is the opposite of the findings of Gundersen et al. [18] in their study involving 450,856 women in Denmark. These authors reported that cesarean sections were associated with a higher risk of urinary infection and the occurrence of related symptoms (stinging feeling and trouble while urinating).

In our study, an instrumental delivery was associated with having hemorrhoids during the puerperium, and a cesarean section was a protective factor against hemorrhoids. Similar results were reported by Ansara et al. [45].

A cesarean section was found to be associated with women suffering back pain during the puerperium, similar to other authors who also found this association [42]. For constipation, we identified its positive association during the puerperium with instrumental delivery. This agrees with the results of Kepenekci et al. [46], who identified vaginal delivery as a risk factor for postpartum constipation. However, this contrasts with the findings by Woolhouse et al., who found no association between mode of delivery and puerperium constipation among 1507 primiparous women in Australia [42].

Healthcare staff must identify those women who desire a cesarean section but have no clinical indication to do so in order to inform them about the consequences that this mode of delivery entails during the puerperium, so that they can make a decision after being properly informed. Furthermore, health policies and future research should focus on the reasons that lead to women choosing a cesarean delivery and developing related strategies, taking into account results such as those in this study to provide adequate information to women about the problems and discomfort associated with cesarean delivery. In the same way, health professionals who attend births should naturally encourage childbirth care, in which the use of delivery instruments is reserved for specific clinical indications, as there is currently an excess of instrumental delivery. The incidence of instrumental deliveries is high in Spain (23%), and some are not justified, as 40% of these are done to prevent problems or for teaching purposes [47].

## 5. Conclusions

In conclusion, an instrumental delivery entails more problems for women during the postpartum period, including constipation, wound infection, perineal pain, hemorrhoids, headache, and psychological alterations, compared with non-instrumental vaginal delivery. Having a cesarean section is associated with a higher probability of wound infection, headache, back pain, maternal feelings of sadness, anxiety and depression, as well as tiredness and symptoms of post-traumatic stress disorder.

## Figures and Tables

**Table 1 ijerph-16-04564-t001:** Characteristics of the women who participated in the study.

Variable	*N* = 3324
	*n* (%)
**Mother’s age**	
<35 years	1664 (50.1)
>35 years	1660 (49.9)
**Level of education**	
No qualifications	8 (0.2)
Primary education	155 (4.7)
Secondary education	1204 (36.2)
University education	1957 (58.9)
**Spanish nationality**	
Yes	3210 (96.6)
**Twin pregnancy**	
Yes	125 (3.8)
**Gestational age**	
Preterm	263 (7.9)
Term	3061 (92.1)
**Attended antenatal classes**	
Yes	2012 (60.5)
**Problems during pregnancy**	
Yes	461 (13.9)
**Parity**	
Primiparous	1725 (51.9)
Multiparous	1599 (48.1)
**Previous Cesarean section**	
Yes	1003 (30.2)
**Induction of Labor**	
Yes	1127 (33.9)
**Use of epidural or spinal anesthesia**	
Yes	2571 (77.3)
**Mode of delivery**	
Normal	1917 (57.7)
Instrumental	603 (18.1)
Cesarean section	804 (24.2)
**Third or fourth degree perineal tears**	
Yes	137 (4.1)
**Episiotomy**	
Yes	1230 (37.0)
**Skin-to-skin**	
Yes	2255 (67.8)
**Newborn hospitalized**	
Yes	280 (8.4)
**Formula feeding upon discharge**	
Yes	972 (29.2)

**Table 2 ijerph-16-04564-t002:** Factors associated with mode of delivery.

Variable	Mode of Delivery		
	Normal *n* (%)	Instrumental *n* (%)	Cesarean *n* (%)
**Mother’s age**			
<35 years	960 (57.7)	326 (19.6)	378 (22.7)
>35 years	957 (57.7)	277 (16.7)	426 (25.7)
OR (95%CI)	1 (ref.)	0.85 (0.71–1.03)	1.13 (0.96–1.33)
**Level of education**			
No qualifications	6 (75.0)	2 (25.0)	0 (0.0)
Primary education	83 (53.5)	34 (21.9)	38 (24.5)
OR (95%CI)	1 (ref.)	1.23 (0.24-6.40)	**NC**
Secondary education	669 (55.6)	246 (20.4)	289 (24.0)
OR (95%CI)	1 (ref.)	1.10 (0.22-5.48)	**NC**
University education	1159 (59.2)	321 (16.4)	477 (24.4)
OR (95%CI)	1 (ref.)	0.83 (0.17-4.12)	**NC**
**Spanish nationality**			
No	1865 (58.1)	569 (17.7)	776 (24.2)
Yes	52 (45.6)	34 (29.8)	28 (24.6)
OR (95%CI)	1 (ref.)	**2.15 (1.38-3.35)**	**1.29 (0.81–2.06)**
**Twin pregnancy**			
No	1892 (59.1)	585 (18.3)	722 (22.6)
Yes	25 (20.0)	18 (14.4)	82 (65.6)
OR (95%CI)	1 (ref.)	**2.34 (1.27–4.32)**	**8.60 (5.45–13.56)**
**Gestational age**			
Term	1802 (58.9)	571 (18.7)	688 (22.5)
Preterm	115 (43.7)	32 (12.2)	116 (44.1)
OR (95%CI)	1 (ref.)	0.87 (0.58–1.31)	**2.64 (2.01–3.47)**
**Problems during pregnancy**			
No	1688 (59.0)	521 (18.2)	654 (22.8)
Yes	229 (49.7)	82 (17.8)	150 (32.5)
OR (95%CI)	1 (ref.)	1.08 (0.75–1.55)	**1.69 (1.35–2.12)**
**Parity**			
Primiparous	784 (45.4)	426 (24.7)	515 (29.9)
Multiparous	1133 (70.9)	177 (11.1)	289 (18.1)
OR (95%CI)	1 (ref.)	**1.30 (1.02–1.66)**	**2.18 (1.73–2.74)**
**Induction of Labor**			
No	1395 (63.5)	375 (17.1)	427 (19.4)
Yes	522 (46.3)	228 (20.2)	377 (33.5)
OR (95%CI)	1 (ref.)	**1.63 (1.34–1.97)**	**2.36 (1.99–2.80)**
**Use of epidural/spinal anesthesia**			
No	656 (87.1)	54 (7.2)	43 (5.7)
Yes	1261 (49.0)	549 (21.4)	761 (29.6)
OR (95%CI)	1 (ref.)	**5.28 (3.93–7.09)**	**9.21 (6.68–12.70)**

NC: not calculated; OR: odds ratio; 95%CI: 95% confidence intervals; (ref.): reference; Bold: Significant results are highlighted.

**Table 3 ijerph-16-04564-t003:** Symptoms of discomfort and self-reported maternal problems at 6 weeks postpartum.

Variable	Mode of Delivery		
	Normal *n* (%)	Instrumental *n* (%)	Cesarean *n* (%)
**Constipation**			
No	1128 (58.8)	287 (47.6)	515 (64.1)
Yes	789 (41.2)	316 (52.4)	289 (35.9)
OR (95%CI)	1 (ref.)	**1.57 (1.31–1.89)**	**0.80 (0.68–0.95)**
aORª (95%CI)	1 (ref.)	**1.35 (1.10–1.66)**	0.90 (0.74–1.09)
**Hemorrhoids**			
No	959 (50.1)	259 (43.0)	570 (70.9)
Yes	958 (49.9)	344 (57.0)	234 (29.1)
OR (95%CI)	1 (ref.)	**1.33 (1.11–1.60)**	**0.41 (0.34–0.49)**
aORª (95%CI)	1 (ref.)	**1.28 (1.04–1.57)**	**0.43 (0.35–0.52)**
**Infected wound**			
No	1868 (97.4)	530 (87.9)	696 (86.6)
Yes	49 (2.6)	73 (12.1)	108 (13.4)
OR (95%CI)	1 (ref.)	**5.25 (3.61–7.64)**	**5.92 (4.17–8.38)**
aOR^b^ (95%CI)	1 (ref.)	**2.45 (1.61–3.71)**	**11.62 (6.77–19.95)**
**Perineal pain**			
No	998 (52.1)	162 (26.9)	734 (91.3)
Yes	919 (47.9)	441 (73.1)	70 (8.7)
OR (95%CI)	1 (ref.)	**2.96 (2.42–3.62)**	**0.10 (0.08–0.13)**
aORª (95%CI)	1 (ref.)	**1.63 (1.30–2.04)**	**0.14 (0.11–0.19)**
**Headache**			
No	1553 (81.0)	447 (74.1)	565 (70.3)
Yes	364 (19.0)	156 (25.9)	239 (29.7)
OR (95%CI)	1 (ref.)	**1.49 (1.20–1.85)**	**1.81 (1.49–2.18)**
aOR^c^ (95%CI)	1 (ref.)	**1.43 (1.13–1.81)**	**1.76 (1.43–2.18)**
**Chest pain**			
No	1100 (57.4)	340 (56.4)	467 (58.1)
Yes	817 (42.6)	263 (43.6)	337 (41.9)
OR (95%CI)	1 (ref.)	1.04 (0.87–1.25)	0.97 (0.82–1.14)
aOR^d^ (95%CI)	1 (ref.)	0.95 (0.79–1.15)	0.96 (0.80–1.14)
**Back pain**			
No	1131 (59.0)	309 (51.2)	406 (50.5)
Yes	786 (41.0)	294 (48.8)	398 (49.5)
OR (95%CI)	1 (ref.)	**1.37 (1.14–1.65)**	**1.41 (1.20–1.66)**
aOR^e^ (95%CI)	1 (ref.)	1.16 (0.95–1.41)	**1.21 (1.01–1.45)**
**Stinging while urinating**			
No	1550 (80.9)	448 (74.3)	705 (87.7)
Yes	367 (19.1)	155 (25.7)	99 (12.3)
OR (95%CI)	1 (ref.)	**1.46 (1.18–1.81)**	**0.59 (0.47–0.75)**
aOR^f^ (95%CI)	1 (ref.)	1.14 (0.89–1.46)	**0.62 (0.47–0.82)**
**Urinary incontinence**			
No	1251 (65.3)	321 (53.2)	677 (84.2)
Yes	666 (34.7)	282 (46.8)	127 (15.8)
OR (95%CI)	1 (ref.)	**1.65 (1.37–1.99)**	**0.35 (0.29–0.44)**
aOR^g^ (95%CI)	1 (ref.)	**1.30 (1.05–1.61)**	**0.32 (0.25–0.42)**
**Fecal incontinence**			
No	1849 (96.5)	533 (88.4)	779 (96.9)
Yes	68 (3.5)	70 (11.6)	25 (3.1)
OR (95%CI)	1 (ref.)	**3.57 (2.52–5.05)**	0.87 (0.54–1.39)
aOR^g^ (95%CI)	1 (ref.)	**1.94 (1.29–2.92)**	1.43 (0.79–2.59)
**Tiredness**			
No	309 (16.1)	83 (13.8)	100 (12.4)
Yes	1608 (83.9)	520 (86.2)	704 (87.6)
OR (95%CI)	1 (ref.)	1.20 (0.93–1.56)	**1.35 (1.06–1.72)**
aOR^h^ (95%CI)	1 (ref.)	0.95 (0.70–1.27)	**1.32 (1.01–1.73)**
**Sadness**			
No	1164 (60.7)	284 (47.1)	376 (46.8)
Yes	753 (39.3)	319 (52.9)	428 (53.2)
OR (95%CI)	1 (ref.)	**1.74 (1.44–2.09)**	**1.76 (1.49–2.08)**
aOR^i^ 95%CI	1 (ref.)	1.11 (0.89–1.37)	**1.31 (1.03–1.68)**
**Anxiety**			
No	1418 (74.0)	368 (61.0)	484 (60.2)
Yes	499 (26.0)	235 (39.0)	320 (39.8)
OR (95%CI)	1 (ref.)	**1.82 (1.50–2.20)**	**1.88 (1.58–2.24)**
aOR^i^ (95%CI)	1 (ref.)	**1.26 (1.01–1.58)**	**1.36 (1.05–1.76)**
**Depression**			
No	1683 (87.8)	482 (79.9)	604 (75.1)
Yes	234 (12.2)	121 (20.1)	200 (24.9)
OR (95%CI)	1 (ref.)	**1.81 (1.42–2.30)**	**2.38 (1.93–2.94)**
aOR^i^ (95%CI)	1 (ref.)	1.07 (0.81–1.43)	**1.67 (1.21–2.34)**
**Symptoms of post-traumatic stress disorder**			
No	1838 (95.9)	496 (82.3)	632 (78.6)
Yes	79 (4.1)	107 (17.7)	172 (21.4)
OR (95%CI)	1 (ref.)	5.02 (3.70–6.82)	6.33 (4.78–8.39)
aOR^i^ (95%CI)	1 (ref.)	2.74 (1.90–3.97)	4.64 (2.94–7.32)
**Sexual problems**			
No	1250 (65.2)	273 (45.3)	498 (61.9)
Yes	667 (34.8)	330 (54.7)	306 (38.1)
OR (95%CI)	1 (ref.)	**2.27 (1.88–2.73)**	1.15 (0.97–1.37)
aOR^j^ (95%CI)	1 (ref.)	**1.39 (1.12–1.72)**	**1.33 (1.08–1.66)**
**Couple having problems**			
No	1378 (71.9)	388 (64.3)	549 (68.3)
Yes	539 (28.1)	215 (35.7)	255 (31.7)
OR (95%CI)	1 (ref.)	**1.42 (1.17–1.72)**	1.19 (0.99–1.42)
aOR^j^ (95%CI)	1 (ref.)	1.03 (0.81–1.27)	1.17 (0.94–1.47)
**Newborn feeding**			
Breastfeeding	1496 (78.0)	419 (69.5)	437 (54.4)
Formula feeding	421 (22.0)	184 (30.5)	367 (45.6)
OR (95%CI)	1 (ref.)	**1.56 (1.27–1.91)**	**2.98 (2.50–3.56)**
aOR^l^ (95%CI)	1 (ref.)	0.97 (0.77–1.23)	**2.16 (1.74–2.69)**
**Lactation problems**			
No	1276 (66.6)	359 (59.5)	453 (56.3)
Yes	641 (33.4)	244 (40.5)	351 (43.7)
OR (95%CI)	1 (ref.)	**1.35 (1.12–1.63)**	**1.54 (1.30–1.83)**
aOR^i^ (95%CI)	1 (ref.)	0.97 (0.77–1.22)	0.92 (0.74–1.14)

aOR: adjusted odds ratio; OR: odds ratio; 95%CI: 95% confidence intervals; (ref.): reference; Bold: Significant results are highlighted. ^a^Adjusted for mother’s age, level of education, mother’s education, parity, twin pregnancy, episiotomy, and third or fourth degree perineal tears. ^b^Adjusted for mother’s age, level of education, mother’s education, parity, episiotomy, third or fourth degree perineal tears, problems during pregnancy, and newborn feeding type. ^c^ Adjusted for mother’s age, level of education, mother’s education, parity, twin pregnancy, use of epidural or spinal anesthesia, problems during pregnancy, and newborn feeding type. ^d^Adjusted for mother’s age, level of education, mother’s education, parity, twin pregnancy, problems during pregnancy, and newborn feeding type. ^e^ Adjusted for mother’s age, level of education, mother’s education, parity, twin pregnancy, problems during pregnancy, use of epidural or spinal anesthesia, gestational age, and newborn feeding type. ^f^ Adjusted for mother’s age, level of education, mother’s education, parity, problems during pregnancy, episiotomy, third or fourth degree perineal tears, use of epidural/spinal anesthesia, gestational age, and newborn feeding type. ^g^ Adjusted for mother’s age, level of education, mother’s education, parity, twin pregnancy, problems during pregnancy, episiotomy, third or fourth degree perineal tears, use of epidural or spinal anesthesia, and gestational age. ^h^ Adjusted for mother’s age, level of education, mother’s education, parity, episiotomy, third or fourth degree perineal tears, problems during pregnancy, newborn feeding type, twin pregnancy, and hospitalized newborn. ^i^ Adjusted for mother’s age, level of education, mother’s education, parity, episiotomy, third or fourth degree perineal tears, problems during pregnancy, induction of labor, use of epidural or spinal anesthesia, newborn feeding type, nationality, gestational age, skin-to-skin, twin pregnancy, and hospitalized newborn. ^j^ Adjusted for mother’s age, level of education, mother’s education, parity, episiotomy, third or fourth degree perineal tears, problems during pregnancy, induction of labor, use of epidural or spinal anesthesia, newborn feeding type, nationality, gestational age, twin pregnancy, and hospitalized newborn. ^l^ Adjusted for mother’s age, level of education, mother’s education, parity, episiotomy, third or fourth degree perineal tears, problems during pregnancy, induction of labor, use of epidural or spinal anesthesia, nationality, gestational age, skin-to-skin, twin pregnancy, and hospitalized newborn.

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
