# Peer review of "Symptoms of Discomfort and Problems Associated with Mode of Delivery During the Puerperium: An Observational Study"

_ijerph, 2019, doi:10.3390/ijerph16224564_

Round 1

Reviewer 1 Report

The manuscript focuses on puerperal problems reported by 3324 women. Authors concentrated on differences in these aspects between 3 groups of mothers in relation to the mode of delivery (i.e. normal vaginal delivery (NVD), instrumental delivery (IVD) or cesarean section (CS) groups). Unfortunately, the study did not include problems during pregnancy, which could be the cause of post-partum complications/symptoms, for example associations between diabetes and infected wound.

There are some limitations and inconsistencies:

1) The authors compared the post-partum problems between NVD and IVD groups as well as between NVD and CS mothers. Please compare maternal reports between the group of IVD and the mothers of CS.

2) How can you explain that nearly half of the studied women were older than 35 years? 

3) Table 2 presents incorrect data - please check "Gestational age" (preterm vs. term deliveries).

4) The authors informed that "women who went to both private and public hospitals" (lines 194-195). What about the ethical approval if: "This study received approval from the Ethics Committee in Clinical Research (CEIC) of the XXX Hospital" (line 109)?

5) The authors wrote that "Those women who gave birth by cesarean section or instrumental birth showed a higher incidence of infection and psychological alterations ...." (abstract and line 173). It is not true, because the study assessed the presence of infection in only one area (i.e. wound caused by birth/during cesarean section).

6) If formula feeding was used in 29.2% of all women (Table 1), please specify this variable in 3 studied groups. Please check the accuracy of lactation problems in Table 3.

7) The authors should precisely present ALL study results. There are serious inconsistencies between Table 3 and "Results" (lines 127-132 (what about perineal pain, headache, anxiety, and sexual problems??); and 161-170 (what about backache, tiredness, and sexual problems), Discussion (please check lines 173-181).

8) The part entitled "Conclusion" is false. The statement that: "... cesarean sections entail more problems for women during the postpartum period, like ... perineal pain and urinary incontinence... than in normal birth." is NOT true for the CS mothers.

9) Please use "Instruction for Authors" for all references.

Author Response

13 November 2019

Dear Editor of the International Journal of Environmental Research and Public Health

Thank you very much for the opportunity to revise and improve the Manuscript ID: ijerph-579499 entitled “Symptoms of discomfort and problems associated with mode of delivery during the puerperium. An observational study”.

A complete account of all points raised by the reviewers is presented in the attached report. All recommendations were taken into consideration and the necessary clarifications appear in the same order. However, if there is need to expand the manuscript to accommodate any further detail, we would be willing to do so.

We would like to thank the reviewers for their time and diligent critique, and we are looking forward to hearing from you at your earliest convenience.

Sincerely,

Juan Miguel Martinez Galiano

Reviewer (1):

Comments and Suggestions for Authors:

English language and style

 (x) Moderate English changes required

Response: Dr Ingrid de Ruiter is a Medical Writer and Public Health Researcher with a background in Clinical Medicine and Public Health. Professional Member of European Medical Writers Association and American Medical Writers Association. She has carefully checked the manuscript. We believe no more grammatical errors appear.

The manuscript focuses on puerperal problems reported by 3324 women. Authors concentrated on differences in these aspects between 3 groups of mothers in relation to the mode of delivery (i.e. normal vaginal delivery (NVD), instrumental delivery (IVD) or cesarean section (CS) groups). Unfortunately, the study did not include problems during pregnancy, which could be the cause of post-partum complications/symptoms, for example associations between diabetes and infected wound.

Response: Thank you for your comments. Diabetes was considered a problem during pregnancy (Table 2). The variable wound infection result is adjusted for the presence of problems during pregnancy (Table 3: bAdjusted for mother’s age, level of education, mother’s education, parity, episiotomy, third/fourth degree perineal tears, problems during pregnancy and lactation type).

Therefore, we believe that the influence of the presence of diabetes on wound infection was minimal.

There are some limitations and inconsistencies:

1) The authors compared the post-partum problems between NVD and IVD groups as well as between NVD and CS mothers. Please compare maternal reports between the group of IVD and the mothers of CS.

Response: Thank you for your comments. The reference group is the one formed by women who had a normal vaginal delivery. Column 3 is the group of women who had a cesarean delivery. The OR expressed in this column is the one that indicates if there is an association with respect to the reference group (women with NVD), as well as if it is positive or negative. At the same time, you can see women who had an instrumental delivery in column 2. You can compare the OR of column 2 and column 3 and see if there are differences between them. It is not pragmatic to construct another table that only compares women with cesarean delivery and women with instrumental delivery. In Table 3, for example, in wound infection it can be seen how there is an increased risk with an OR of 2.45 for IVD compared with NVD, which jumps up to an OR of 11.62 for women with a caesarean compared to an NVD. It can also be seen how the risk of fecal incontinence is associated with instrumental delivery but not with cesarean delivery.

2) How can you explain that nearly half of the studied women were older than 35 years?

Response: Thank you for your comments. In Spain the age to be a mother for the first time is around 31-32 years [Molina-García L, Hidalgo-Ruiz M, Arredondo-López B, Colomino-Ceprián S, Delgado-Rodríguez M, Martínez-Galiano JM. Maternal Age and Pregnancy, Childbirth and the Puerperium: Obstetric Results. J Clin Med. 2019;8(5):672. Published 2019 May 13. doi:10.3390/jcm8050672; 15.        Instituto Nacional de Estadística. Demografía y población: Encuesta de Fecundidad. 2018 Available at: https://www.ine.es/dyngs/INEbase/es/operacion.htm?c=Estadistica_C&cid=1254736177006&menu=ultiDatos&idp=1254735573002 (Accessed: 1 March 2019).].

Half of our sample is multiparous therefore this can explain why almost half of the sample is 35 years old: already had a previous child at the age of 31-32 years and these women have become a mother again 2-3 years after they were mothers for the first time. In addition, our results are representative of the reference population (Data from the National Statistics Institute of Spain): Mother’s age < 35 years: 262052 (62.3%) 

 and mother´s age ≥ 35 years: 158238 (37.6 %).

3) Table 2 presents incorrect data - please check "Gestational age" (preterm vs. term deliveries).

Response: Thank you for spotting this, it was a typo. We have corrected this.

4) The authors informed that "women who went to both private and public hospitals" (lines 194-195). What about the ethical approval if: "This study received approval from the Ethics Committee in Clinical Research (CEIC) of the XXX Hospital" (line 109)?

Response: Thank you for your comments. In Spain, it is sufficient to obtain a favorable report from an ethics committee. This favorable report is extended to be able to do the study throughout the Spanish territory.

5) The authors wrote that "Those women who gave birth by cesarean section or instrumental birth showed a higher incidence of infection and psychological alterations ...." (abstract and line 173). It is not true, because the study assessed the presence of infection in only one area (i.e. wound caused by birth/during cesarean section).

Response: Thank you for your comments. We have now specified this as “wound infection” for clarity.

6) If formula feeding was used in 29.2% of all women (Table 1), please specify this variable in 3 studied groups. Please check the accuracy of lactation problems in Table 3.

Response: Thank you for your comments. We have included newborn feeding type in Table 3, methods and results.

7) The authors should precisely present ALL study results. There are serious inconsistencies between Table 3 and "Results" (lines 127-132 (what about perineal pain, headache, anxiety, and sexual problems??); and 161-170 (what about backache, tiredness, and sexual problems), Discussion (please check lines 173-181).

Response: Thank you for your comments. To avoid duplication of data, only some of the results are expressed in the text and the reader referred to the table in order to check all the results. This facilitates reading for the reader. We have checked the text and corrected it.

8) The part entitled "Conclusion" is false. The statement that: "... cesarean sections entail more problems for women during the postpartum period, like ... perineal pain and urinary incontinence... than in normal birth." is NOT true for the CS mothers.

Response: Thank you for your comments.  We have checked the text and revised it.

9) Please use "Instruction for Authors" for all references.

Response: Thank you for your comments. We have checked the text and corrected it.

Reviewer 2 Report

Manuscript ijerph-579499

 Title: Symptoms of discomfort and problems associated with mode of delivery during the puerperium. An observational study

Reviewer’s Comments

Most search engines, websites or databases use the words found in the title, abstract, and keywords to display the paper when someone searches with a keyword. Thus, the title of the paper, keywords and the abstract decide whether the paper will be found and cited by another researcher or reader. Here are my suggestions to the authors:

Title

Change full stop after “puerperium to colon. The colon is used to separate two independent clauses when the second explains or illustrates the first. Do not capitalize the first word after the colon as this word is not ordinarily capitalized.

Suggested title: Symptoms of discomfort and problems associated with mode of delivery during the puerperium: an observational study

 Abstract

The Abstract conveys a research gap in the literature, the need, and the importance of this study and the findings of the study have the potential to inform policy changes. The English wording could however be improved; I suggest rewording the Abstract along these lines:

Suggested Abstract

In spite of abundant literature on antenatal and delivery care received by pregnant women, there is a wide knowledge gap on the prevalence of symptoms of discomfort and/or problems during the postpartum period and their relationship to the mode of delivery. This cross-sectional study carried out in Spain in 2017 with 3,324 participants aimed to investigate the association between mode of delivery and the postpartum symptoms of discomfort and maternal problems self-reported by women during puerperium. An ad hoc online questionnaire was used to collect data on socio-demographic and obstetric variables, symptoms of discomfort and maternal problems during the puerperium. The crude odds ratios (OR) and adjusted OR (aOR) and their 95% confidence intervals (95%CI) were calculated by binary logistic regression. 3,324 women participated. Cesarean section vs normal vaginal birth showed an association with infected surgical wound (aOR=11.62, 95%CI: 6.77-19.95), feeling sad (aOR=1.31, 23 95%CI: 1.03-1.68) and symptoms of posttraumatic stress (aOR=4.64, 95%CI: 2.94-7.32). Instrumental birth vs normal vaginal birth was a risk factor by presenting constipation (aOR=1.35 95%CI: 25 1.10-1.66), haemorrhoids (aOR=1.28, 95%CI: 1.04-1.57), urinary incontinence (aOR=1.30, 95%CI: 26 1.05-1.61) and fecal incontinence (aOR=1.94, 95%CI: 1.29-2.92) during the puerperium. Women who gave birth by cesarean section or instrumental birth showed a higher incidence of infection and psychological alterations than those who gave birth vaginally. Identifying women at risk of giving birth by cesarean section and informing them about subsequent symptoms of discomfort and maternal problems during the puerperium must be included in pregnancy health programme policies and protocols to allow women make informed decisions regarding their birthing plan.

Keywords:  I suggest including “birthing plan” and “policy” in your keywords

Keywords: mode of delivery; puerperium; health problems; discomfort; maternal problems; self-perceived health; birthing plan; policy

English language

The manuscript needs a major revision of the language for more adequate sentence structures. Below are a few examples:

The term “cesarean section” needs to be used consistently. While the manuscript uses mostly the non-capitalized American English form– “Cesarean section” – it also uses the capitalized “Cesarean section” form within a sentence.  

Line 48 - with forceps or non-programmed cesarean section presented a worse postpartum psychology health than those who had normal vaginal births or programmed cesarean sections

“psychology health” should be changed to either psychological health or mental health.

Line 51 - studies that included a total population of 2,851 mujeres

Change mujeres to women

Line 59- The effect of mode of delivery that women undergo on different health parameters has been studied, but with inconsistent results.

Change to: “The effect of the mode of delivery on different postpartum health indicators has been studied, but with inconsistent results”. Also, this needs a citation.

Line 180 - We also identified an instrumental birth vs normal birth as a risk factor of feeling a stinging feeling 180 while urinating and urinary/fecal incontinence.

Change to: We also identified instrumental birth vs normal birth as a risk factor for a stinging feeling while urinating and for urinary and/or fecal incontinence.

There are many grammar issues that need to be addressed; I gave you only a few examples.

 Introduction

This section does not adequately describe the status quo in Spain (policies and protocols on prenatal informed decision-making by pregnant women) nor the need and the importance of this study to inform policy changes. In short, it does not support the Abstract.

To inform the reader in this section, you need to provide not only research findings from China, Lebanon, and Jordan, but also from the European Union in general, and from Spain, in special. Have there been in EU any multi-country surveys on correlates of delivery mode with puerperal health, the way for example the 2005 Euro‐Peristat project (25 EU states, plus Norway) collected data on variations between high‐income countries in their rates of obstetric intervention?  If yes, cite; if not emphasize that research gap.

Your references have only one (1) study conducted in Spain; reference nr 5.

Navarro P, García-Esteve L, Ascaso C, Aguado J, Gelabert E, Martín-Santos R. Non-psychotic psychiatric 273 disorders after childbirth: Prevalence and comorbidity in a community sample. J Affect Disord [Internet]. 274 2008 [cited 2018 Jul 11];109:171–6. Available from: http://www.ncbi.nlm.nih.gov/pubmed/18001842

Are there any other relevant studies conducted in Spain? If yes, cite; if not emphasize that research gap.

Make a stronger case why your study was needed. For example, Sword et al (2011) study in Norway suggested that the delivery mode had no significant impact on the development of postpartum depression in the main‐effects model. However, it may interact with place of birth and other unmeasured factors to create a risk for depression.

Sword W, Kurtz Landy C, Thabane L, Watt S, Krueger P, Farine D, Foster G. Is mode of delivery associated with postpartum depression at 6 weeks: a prospective cohort study. BJOG 2011;118:966–977.

Perhaps you could cite this and say that your study intends to explore and verify this finding.

It would also help the reader to see in Introduction some difference between between Spanish and migrant pregnant women

Bernis, C; Varea, C; Gonzales, AE (2012). Labor Management and Mode of Delivery Among Migrant and Spanish Women: Does the Variability Reflect Differences in Obstetric Decisions According to Ethnic Origin? Maternal and Child Health Journal, 17, 918-927

Material and methods

Adequate

Results

Adequate

Discussion

You make a case against selection, information, amnesic, and residual confounding biases; are there any limitations to your study? Are there any results you have not reported? 

Conclusions

Focused but too brief (telegraphic) and incomplete. Where do you see the direction of future, needed research?

Author Response

13 November 2019

Dear Editor of the International Journal of Environmental Research and Public Health

Thank you very much for the opportunity to revise and improve the Manuscript ID: ijerph-579499 entitled “Symptoms of discomfort and problems associated with mode of delivery during the puerperium. An observational study”.

A complete account of all points raised by the reviewers is presented in the attached report. All recommendations were taken into consideration and the necessary clarifications appear in the same order. However, if there is need to expand the manuscript to accommodate any further detail, we would be willing to do so.

We would like to thank the reviewers for their time and diligent critique, and we are looking forward to hearing from you at your earliest convenience.

Sincerely,

Juan Miguel Martinez Galiano

Reviewer (2):

Comments and Suggestions for Authors:

English language and style

(x) Extensive editing of English language and style required 

Response: Dr Ingrid de Ruiter is a Medical Writer and Public Health Researcher with a background in Clinical Medicine and Public Health. Professional Member of European Medical Writers Association and American Medical Writers Association. She has carefully checked the manuscript. We believe no more grammatical errors appear.

Title: Symptoms of discomfort and problems associated with mode of delivery during the puerperium. An observational study Title: Symptoms of discomfort and problems associated with mode of delivery during the puerperium. An observational study

Reviewer’s Comments

Most search engines, websites or databases use the words found in the title, abstract, and keywords to display the paper when someone searches with a keyword. Thus, the title of the paper, keywords and the abstract decide whether the paper will be found and cited by another researcher or reader. Here are my suggestions to the authors:

Title

Change full stop after “puerperium to colon. The colon is used to separate two independent clauses when the second explains or illustrates the first. Do not capitalize the first word after the colon as this word is not ordinarily capitalized.

Suggested title: Symptoms of discomfort and problems associated with mode of delivery during the puerperium: an observational study

Response: Thank you for your comments. We have corrected it.

 Abstract

The Abstract conveys a research gap in the literature, the need, and the importance of this study and the findings of the study have the potential to inform policy changes. The English wording could however be improved; I suggest rewording the Abstract along these lines:

Suggested Abstract

In spite of abundant literature on antenatal and delivery care received by pregnant women, there is a wide knowledge gap on the prevalence of symptoms of discomfort and/or problems during the postpartum period and their relationship to the mode of delivery. This cross-sectional study carried out in Spain in 2017 with 3,324 participants aimed to investigate the association between mode of delivery and the postpartum symptoms of discomfort and maternal problems self-reported by women during puerperium. An ad hoc online questionnaire was used to collect data on socio-demographic and obstetric variables, symptoms of discomfort and maternal problems during the puerperium. The crude odds ratios (OR) and adjusted OR (aOR) and their 95% confidence intervals (95%CI) were calculated by binary logistic regression. 3,324 women participated. Cesarean section vs normal vaginal birth showed an association with infected surgical wound (aOR=11.62, 95%CI: 6.77-19.95), feeling sad (aOR=1.31, 23 95%CI: 1.03-1.68) and symptoms of posttraumatic stress (aOR=4.64, 95%CI: 2.94-7.32). Instrumental birth vs normal vaginal birth was a risk factor by presenting constipation (aOR=1.35 95%CI: 25 1.10-1.66), haemorrhoids (aOR=1.28, 95%CI: 1.04-1.57), urinary incontinence (aOR=1.30, 95%CI: 26 1.05-1.61) and fecal incontinence (aOR=1.94, 95%CI: 1.29-2.92) during the puerperium. Women who gave birth by cesarean section or instrumental birth showed a higher incidence of infection and psychological alterations than those who gave birth vaginally. Identifying women at risk of giving birth by cesarean section and informing them about subsequent symptoms of discomfort and maternal problems during the puerperium must be included in pregnancy health programme policies and protocols to allow women make informed decisions regarding their birthing plan.

Response: Thank you for your comments. We have revised it as recommended.

Keywords:  I suggest including “birthing plan” and “policy” in your keywords

Keywords: mode of delivery; puerperium; health problems; discomfort; maternal problems; self-perceived health; birthing plan; policy

Response: Thank you for your comments. We have added these as suggested.

English language

The manuscript needs a major revision of the language for more adequate sentence structures. Below are a few examples:

The term “cesarean section” needs to be used consistently. While the manuscript uses mostly the non-capitalized American English form– “Cesarean section” – it also uses the capitalized “Cesarean section” form within a sentence.

Response: Dr Ingrid de Ruiter is a Medical Writer and Public Health Researcher with a background in Clinical Medicine and Public Health. Professional Member of European Medical Writers Association and American Medical Writers Association. She has carefully checked the manuscript. We believe no more grammatical errors appear.

Line 48 - with forceps or non-programmed cesarean section presented a worse postpartum psychology health than those who had normal vaginal births or programmed cesarean sections

“psychology health” should be changed to either psychological health or mental health.

Response: Thank you for your comments. We have corrected this.

Line 51 - studies that included a total population of 2,851 mujeres

Change mujeres to women

Response: Thank you for your comments. We have corrected it.

Line 59- The effect of mode of delivery that women undergo on different health parameters has been studied, but with inconsistent results.

Change to: “The effect of the mode of delivery on different postpartum health indicators has been studied, but with inconsistent results”. Also, this needs a citation.

Response: Thank you for your comments. We have corrected this and have added the references.

Line 180 - We also identified an instrumental birth vs normal birth as a risk factor of feeling a stinging feeling 180 while urinating and urinary/fecal incontinence.

Change to: We also identified instrumental birth vs normal birth as a risk factor for a stinging feeling while urinating and for urinary and/or fecal incontinence.

There are many grammar issues that need to be addressed; I gave you only a few examples.

Response: Dr Ingrid de Ruiter is a Medical Writer and Public Health Researcher with a background in Clinical Medicine and Public Health. Professional Member of European Medical Writers Association and American Medical Writers Association. She has carefully checked the manuscript. We believe no more grammatical errors appear.

Introduction

This section does not adequately describe the status quo in Spain (policies and protocols on prenatal informed decision-making by pregnant women) nor the need and the importance of this study to inform policy changes. In short, it does not support the Abstract.

To inform the reader in this section, you need to provide not only research findings from China, Lebanon, and Jordan, but also from the European Union in general, and from Spain, in special. Have there been in EU any multi-country surveys on correlates of delivery mode with puerperal health, the way for example the 2005 Euro‐Peristat project (25 EU states, plus Norway) collected data on variations between high‐income countries in their rates of obstetric intervention?  If yes, cite; if not emphasize that research gap.

Your references have only one (1) study conducted in Spain; reference nr 5.

Navarro P, García-Esteve L, Ascaso C, Aguado J, Gelabert E, Martín-Santos R. Non-psychotic psychiatric 273 disorders after childbirth: Prevalence and comorbidity in a community sample. J Affect Disord [Internet]. 274 2008 [cited 2018 Jul 11];109:171–6. Available from: http://www.ncbi.nlm.nih.gov/pubmed/18001842

Are there any other relevant studies conducted in Spain? If yes, cite; if not emphasize that research gap.

Make a stronger case why your study was needed. For example, Sword et al (2011) study in Norway suggested that the delivery mode had no significant impact on the development of postpartum depression in the main‐effects model. However, it may interact with place of birth and other unmeasured factors to create a risk for depression.

Sword W, Kurtz Landy C, Thabane L, Watt S, Krueger P, Farine D, Foster G. Is mode of delivery associated with postpartum depression at 6 weeks: a prospective cohort study. BJOG 2011;118:966–977.

Perhaps you could cite this and say that your study intends to explore and verify this finding.

It would also help the reader to see in Introduction some difference between between Spanish and migrant pregnant women

Bernis, C; Varea, C; Gonzales, AE (2012). Labor Management and Mode of Delivery Among Migrant and Spanish Women: Does the Variability Reflect Differences in Obstetric Decisions According to Ethnic Origin? Maternal and Child Health Journal, 17, 918-927

Response: Thank you for your comments. The number of immigrant women in the study is low (less than 4%), therefore we believe that addressing this aspect is not relevant. However, we have introduced the suggested reference [Sword W, Kurtz Landy C, Thabane L, Watt S, Krueger P, Farine D, Foster G. Is mode of delivery associated with postpartum depression at 6 weeks: a prospective cohort study. BJOG 2011;118:966–977.]. In terms of other relevant studies in Spain, we have only identified that one studycited. We have rewritten the introduction with the suggestions made.

Material and methods

Adequate

Response: Thank you for your comments

Results

Adequate

Response: Thank you for your comments

Discussion

You make a case against selection, information, amnesic, and residual confounding biases; are there any limitations to your study? Are there any results you have not reported? 

Response: Thank you for your comments. All studies have strengths and limitations. These are the possible limitations and the way in which they were addressed in order to reduce their impact or to compensate. The following are the limitations: “If a selection bias was actually associated with non-response, it did not affect our results. The majority of the women responded positively to participate and only 29 refused. There is nothing to suggest that the non-responding women would have acted differently from those who did. It is unlikely that an information bias exists: the collected data and the way answers were presented in such a way that anyone with any level of education could understand them as they were simply presented. We cannot exclude an amnesic bias, although information was collected over a short period. Therefore, if this bias had any influence on the results, we believe it would be weak. Women were perfectly aware who their healthcare supplier was and if they went to a private or a public hospital. They remembered details about their birth process, and most paid attention and evaluated the whole process very well. We cannot rule out a residual confounding bias, even though any influence on the results would have been minimal as fitting each variable was done individually and specifically for those women who it could have influenced.”

The results are part of a larger study, however all the results related to the discomfort and problems associated with the type of delivery are presented in this article.

Conclusions

Focused but too brief (telegraphic) and incomplete. Where do you see the direction of future, needed research?

Response: Thank you for your comments.  We have rewritten the conclusion and have added a paragraph with the future implications that could be carried out in terms of both health policies and future research in discussion section.

Reviewer 3 Report

Referee Report for IJERPH
“Symptoms of discomfort and problems associated with mode of delivery during the puerperium. An
observational study”
Manuscript ID: ijerph-579499
Synopsis:
This paper analyzes the association of mode of delivery (normal vaginal birth, instrument-assisted birth,
or c-section) with immediate postpartum outcomes, such as infection, incontinence, hemorrhoids,
depression, anxiety, trouble breastfeeding, and PTSD among others. To study this relationship, the
authors developed an online questionnaire and administered it to 3,324 women.
I found this paper interesting and timely, and overall, I support publication. However, the paper has a
number of English-language errors, and is lacking in the written and statistical presentation. In
particular, the introduction and discussion sections can be strengthened. I also have a few questions
about the statistical analyses. I believe a major re-write is needed before the manuscript is suitable for
publication.
Main Comments:
(1) As mentioned, the writing has extensive errors. I would suggest finding a native-language editor
to go through thoroughly and help the authors revise the writing. Some examples (and these are
just a few from the first page or so) include:
a. Lines 16-20: Determine the association between mode of delivery and the different
symptoms of discomfort and maternal problems self-reported by women during the
postpartum period. A cross-sectional study with puerperal women in Spain to collect
data on socio-demographic and obstetric variables, symptoms of discomfort and
maternal problems during the puerperium.
These are not full sentences.
b. Lines 40-41: Some factors related with pregnancy and birth exist, as do certain clinical
practices carried out while attending birth, which have been associated with greater
maternal morbidity.
Not correct English; unclear intention and meaning of sentences.
c. Line 110: This study received approval from the Ethics Committee in Clinical Research
(CEIC) of the XXX Hospital.
“XXX” is a typo
d. Lines 81-82: …the Spanish Federation of Midwives Associations (FAME), its member
associations and midwives to diffuse the project and encourage women to participate.
‘Diffuse’ is wrong word choice.
e. Line 118: This table also shows how most women had a low-risk pregnancy; that is, they
had no health problems during pregnancy: 86.1% (2,863 women).
Should be “This table also shows how most women (2,863, or 86.1%) had a low-risk
pregnancy; that is, they had no health problems during pregnancy.”
f. Lines 98-99: The independent variables considered to control confounder factors were of
the clinical and socio-demographic kind. For each outcome, the variables that could act
as confounders were used.
Better would be “We used a variety of clinical and socio-demographic variables as
controls in each of the bivariate analyses.”
(2) Introduction – Your introduction section needs to better set-up your punchline – namely (as I
understand it), that there is an overuse of patient-requested/medically unnecessary c-section in
Spain and that the negative outcomes you will show are associated with c-sections can be
reduced if there were fewer c-sections. However, you don’t explicitly lay that out for the reader.
The paragraph on lines 55-58 is an attempt, but is not well-integrated into the story line and is
missing the point. In addition, more literature about that issue would be helpful.
(3) The methods/statistics generally looks fine. However, I have a few comments:
a. In the discussion, the authors mention (line 183) that “the majority of women
responded positively to participate and only 29 refused.” More details about this should
be provided in the Methods section. Currently, the authors mention that FAME helped
recruit participants, but from where? All hospitals in spain? Are there home births? If so,
were those women included? How did the recruiting happen? Via email? In person?
What were the exact #s of women approached? How many refused? Were any
respondents excluded after participation? Did all fill in all questions? Etc
b. I am surprised the authors did not include a variable about birth attendant, in particular
midwife vs obstetrician. Do you have this variable? If so, why not include it as a
cofounder in the adjusted ORs from Table 3? (And it certainly should be in Table 2, but I
now discuss that table separately…)
c. Table 2 – factors associated with mode of delivery – not sure if this is relevant to your
study, which is about the correlation between mode of delivery and outcomes. While
interesting, it distracts a little as it presents its own problems. For instance, you look at
the relationship between parity and mode of delivery, but are you controlling for
whether the multiparous women’s previous birth was by c-section? I would recommend
putting this Table in appendix and referencing it there if necessary.
d. Tables 1 & 3. Should be condensed to use more columns and fewer rows – too long and
hard to read.
e. Table 3.
i. The notes under table 3 can be greatly condensed, or at least remove the
carriage lines and reduce the font.
ii. Also, explanation is needed about why the confounders vary across outcomes.
(4) Discussion –
a. You need to discuss limitations of your study
b. I think the paragraph currently in the conclusion should be moved to the discussion
section and expanded upon, as this is your main point. As I understand it, you are
basically suggesting that your results (which show a lot of negative outcomes from csections) can be used as a way to discourage women who request c-sections, but who
show no medical indication for one. If that’s the case, you need to be more explicit
about this and explain how it could happen, etc. The conclusion has other purposes.
c. In addition, you gloss over the findings of instrument-assisted births, which also have
negative outcomes associated with them. Is there a way to reduce these? Are they
usually medically necessary/indicated in Spain, or are they also overused as in the US?

Author Response

13 November 2019

Dear Editor of the International Journal of Environmental Research and Public Health

Thank you very much for the opportunity to revise and improve the Manuscript ID: ijerph-579499 entitled “Symptoms of discomfort and problems associated with mode of delivery during the puerperium. An observational study”.

A complete account of all points raised by the reviewers is presented in the attached report. All recommendations were taken into consideration and the necessary clarifications appear in the same order. However, if there is need to expand the manuscript to accommodate any further detail, we would be willing to do so.

We would like to thank the reviewers for their time and diligent critique, and we are looking forward to hearing from you at your earliest convenience.

Sincerely,

Juan Miguel Martinez Galiano

Reviewer (3):

Comments and Suggestions for Authors:

English language and style

(x) Extensive editing of English language and style required

Response: Dr Ingrid de Ruiter is a Medical Writer and Public Health Researcher with a background in Clinical Medicine and Public Health. Professional Member of European Medical Writers Association and American Medical Writers Association. She has carefully checked the manuscript. We believe no more grammatical errors appear.

Referee Report for IJERPH

“Symptoms of discomfort and problems associated with mode of delivery during the puerperium. An observational study”

Manuscript ID: ijerph-579499

Synopsis:

This paper analyzes the association of mode of delivery (normal vaginal birth, instrument-assisted birth, or c-section) with immediate postpartum outcomes, such as infection, incontinence, hemorrhoids, depression, anxiety, trouble breastfeeding, and PTSD among others. To study this relationship, the authors developed an online questionnaire and administered it to 3,324 women.

I found this paper interesting and timely, and overall, I support publication. However, the paper has a number of English-language errors, and is lacking in the written and statistical presentation. In particular, the introduction and discussion sections can be strengthened. I also have a few questions about the statistical analyses. I believe a major re-write is needed before the manuscript is suitable for publication.

Response: Thank you for your comments.

Main Comments:

(1) As mentioned, the writing has extensive errors. I would suggest finding a native-language editor to go through thoroughly and help the authors revise the writing. Some examples (and these arejust a few from the first page or so) include:

Lines 16-20: Determine the association between mode of delivery and the different symptoms of discomfort and maternal problems self-reported by women during the postpartum period. A cross-sectional study with puerperal women in Spain to collect data on socio-demographic and obstetric variables, symptoms of discomfort and maternal problems during the puerperium. These are not full sentences.

Response: Dr Ingrid de Ruiter is a Medical Writer and Public Health Researcher with a background in Clinical Medicine and Public Health. Professional Member of European Medical Writers Association and American Medical Writers Association. She has carefully checked the manuscript. We believe no more grammatical errors appear.

Lines 40-41: Some factors related with pregnancy and birth exist, as do certain clinical practices carried out while attending birth, which have been associated with greater maternal morbidity.

Not correct English; unclear intention and meaning of sentences.

Response: Dr Ingrid de Ruiter is a Medical Writer and Public Health Researcher with a background in Clinical Medicine and Public Health. Professional Member of European Medical Writers Association and American Medical Writers Association. She has carefully checked the manuscript. We believe no more grammatical errors appear.

Line 110: This study received approval from the Ethics Committee in Clinical Research (CEIC) of the XXX Hospital. “XXX” is a typo

Response: Thank you for spotting this. We have corrected this: “This study received approval from the Ethics Committee in Clinical Research (CEIC) of the La Mancha-Centro Hospital with ethical code 69-C.”

Lines 81-82: …the Spanish Federation of Midwives Associations (FAME), its member associations and midwives to diffuse the project and encourage women to participate. ‘Diffuse’ is wrong word choice.

Response: Thank you for your comments. We have corrected it and revised this sentence.

Line 118: This table also shows how most women had a low-risk pregnancy; that is, they had no health problems during pregnancy: 86.1% (2,863 women). Should be “This table also shows how most women (2,863, or 86.1%) had a low-risk pregnancy; that is, they had no health problems during pregnancy.”

Response: Thank you for your comments. We have corrected it: “This table also shows how most women (86.1%) had a low-risk pregnancy; that is, they had no health problems during”

Lines 98-99: The independent variables considered to control confounder factors were of the clinical and socio-demographic kind. For each outcome, the variables that could act as confounders were used. Better would be “We used a variety of clinical and socio-demographic variables as controls in each of the bivariate analyses.”

Response: Thank you for your comments. We have corrected it: “We used a variety of clinical and socio-demographic variables as controls in each of the bivariate analyses”

(2) Introduction – Your introduction section needs to better set-up your punchline – namely (as I understand it), that there is an overuse of patient-requested/medically unnecessary c-section in Spain and that the negative outcomes you will show are associated with c-sections can be reduced if there were fewer c-sections. However, you don’t explicitly lay that out for the reader.

The paragraph on lines 55-58 is an attempt, but is not well-integrated into the story line and is missing the point. In addition, more literature about that issue would be helpful.

Response: Thank you very much for your comments. We have rewritten the paragraph and have introduced some references “Sometimes women request giving birth by a specific mode of delivery. Some prefer giving birth as naturally as possible with no medication, as set out by the World Health Organization (WHO) [19]. However, more women request giving birth by cesarean section with no clinical indication to justify this decision other than them not wishing to undergo vaginal birth [20–24]. Among women with a caesarean, between 1–48% in countries like the United States and 18% in the UK was at the request of the mother herself [Karlström A, Nystedt A, Johansson M, Hildingsson I. Behind the myth--few women prefer caesarean section in the absence of medical or obstetrical factors. Midwifery. 2011 Oct;27(5):620-7. doi: 10.1016/j.midw.2010.05.005. Epub 2010 Jul 13.] In Spain, there are also a considerable number of caesarean sections performed without clinical indication; rather among the reasons is the mother's request. [Márquez-Calderón S, Ruiz-Ramos M, Juárez S, Librero López J. Caesarean delivery in Andalusia, Spain: relationship with social, clinical and health services factors (2007-2009)] Rev Esp Salud Publica. 2011 Mar-Apr;85(2):205-15. doi: 10.1590/S1135-57272011000200008] Do women really know the risks, symptoms of discomfort, and problems that one type of birth or another can entail?

(3) The methods/statistics generally looks fine. However, I have a few comments:

In the discussion, the authors mention (line 183) that “the majority of women responded positively to participate and only 29 refused.” More details about this should be provided in the Methods section. Currently, the authors mention that FAME helped recruit participants, but from where? All hospitals in spain? Are there home births? If so, were those women included? How did the recruiting happen? Via email? In person? What were the exact #s of women approached? How many refused? Were any respondents excluded after participation? Did all fill in all questions? Etc

Response: Thank you very much for your comments.  In Spain, births at home are scarce and are not contemplated by health providers (neither public nor private). Those who attend this type of birth are midwives who practice independently. Therefore, we do not have any home deliveries in our study. The women who participated, completed the questionnaire completely. The women who refused did so by proposing to participate in the study. The FAME sent an email to the midwives and from their work centers these midwives recruited women in different areas of Spain. The women were fully informed, and an address was provided to complete the questionnaire. We have introduced some clarifications in the text.

I am surprised the authors did not include a variable about birth attendant, in particular midwife vs obstetrician. Do you have this variable? If so, why not include it as a cofounder in the adjusted ORs from Table 3? (And it certainly should be in Table 2, but I now discuss that table separately…)

Response:  Thanks for your comment. We did not specifically collect the professional who attended the delivery. Although, in Spain all normal deliveries are attended by midwives while instrumental deliveries and cesarean sections are attended by gynecologists. No other professional can attend births legally in Spain.

Table 2 – factors associated with mode of delivery – not sure if this is relevant to your study, which is about the correlation between mode of delivery and outcomes. While interesting, it distracts a little as it presents its own problems. For instance, you look at the relationship between parity and mode of delivery, but are you controlling for whether the multiparous women’s previous birth was by c-section? I would recommend putting this Table in appendix and referencing it there if necessary.

Response:  Thanks for your comment. As you say, the table is interesting. Moving to an appendix may lose its interest and make reading difficult for the reader.

Tables 1 & 3. Should be condensed to use more columns and fewer rows – too long and hard to read.

Response:  Thank you for your comments. We are sorry. We are aware of the complexity of the tables; however, we want to provide all the information. We have not found another style to present the results.

Table 3. The notes under table 3 can be greatly condensed, or at least remove the carriage lines and reduce the font.

Response:  Thank you for your comments. We send the document in word format. The editorial office of the journal establishes the format and the font. We cannot modify the norms of the journal.

Also, explanation is needed about why the confounders vary across outcomes.

Response:  Thank you for your comments. We have rewritten the discussion section to clarify “The women who agreed to participate filled in all the questions completely. We cannot rule out a residual confounding bias, even though any influence on the results would have been minimal as fitting each result variable was done individually and specifically for result was adjusted specifically for all possible variables that could have influenced on them. These variables were determined based on those found in the literature review and the clinical experience of the researchers.”

(4) Discussion –

You need to discuss limitations of your study

Response:  Thank you for your comments. All studies have strengths and limitations. These are the possible limitations and the way in which it has been addressed. We have rewritten the discussion section to clarify: “If a selection bias was actually associated with non-response, it did not affect our results. The majority of the women responded positively to participate and only 29 refused. There is nothing to suggest that the non-responding women would have acted differently from those who did. It is unlikely that an information bias exists: the collected data and the way answers were presented in such a way that anyone with any level of education could understand them as they were simply presented. We cannot exclude a memory bias, although information was collected over a short period. So, if this bias had any influence on the results, we believe it would be weak. Women knew their healthcare supplier and if they went to a private or a public hospital. They remembered details about their birth process, and most paid attention and evaluated the whole process very well. We cannot rule out a residual confounding bias, even though any influence on the results would have been minimal as fitting each variable was done individually and specifically for those women who it could have influenced.”

I think the paragraph currently in the conclusion should be moved to the discussion section and expanded upon, as this is your main point. As I understand it, you are basically suggesting that your results (which show a lot of negative outcomes from csections) can be used as a way to discourage women who request c-sections, but who show no medical indication for one. If that’s the case, you need to be more explicit about this and explain how it could happen, etc. The conclusion has other purposes.

Response:  Thank you for your comments. We have rewritten the conclusion and we have moved to the discussion section the future implications for health policy and future research lines. “CONCLUSION:   By way of conclusion, instrumental birth entails more problems for women during the postpartum period, including constipation, wound infection, perineal pain, hemorrhoids, headache, and psychological alterations compared with non-instrumental vaginal birth. Having a cesarean section is associated with a higher probability of wound infection, headache, back pain, maternal feelings of sadness, anxiety and depression, as well as tiredness and symptoms of posttraumatic stress disorder.

In addition, you gloss over the findings of instrument-assisted births, which also have negative outcomes associated with them. Is there a way to reduce these? Are they usually medically necessary/indicated in Spain, or are they also overused as in the US?

Response:  Thank you for your comments. We have added a paragraph. “In the same way, health professionals who attend births should naturally opt for delivery assistance by reserving the use of delivery instruments for specific indications: there is currently an excess of instrumental delivery. The incidence of instrumental deliveries is high in Spain (23%), and the reason for some is not well justified, as 40% of these is done to prevent problems or for teaching purposes [Aceituno-Velasco L. Tasa de partos instrumentales en España. Prog Obstet Ginecol. 2009;52(10):609-15].”

Reviewer 4 Report

        Thank you for the invitation to review the article entitled “Symptoms of discomfort and problems associated with mode of delivery during the puerperium. An observational study” (ijerph-579499). The article aimed to determine the association between mode of delivery and the different symptoms of discomfort and maternal problems self-reported by women during the postpartum period using a cross-sectional survey study. My comments are listed below:

The authors may try to avoid multiple citations in one single sentence, especially in the section of Introduction. In the method, the authors stated that “we conducted a cross-sectional study with the women attended to while giving birth in 2017 in Spain Women aged under 18 and the births that ended in fetal death antenatally were excluded.” This sentence is very confusing. Does the “Spain Women” mean Spanish women or refer to an organization or a hospital? The authors mean “…while giving birth in 2017 in Spain Women aged under 18 “, or “aged under 18 and the births that ended in fetal death antenatally were excluded”? In the paragraph of Ethical approval, the authors should specify the hospital that granted them the IRB approval, not just using XXX. Table 1 consists of many Yes/No questions. For those questions, data of either “Yes” or “No” would be sufficient. The layout of Table 2 or 3 could be modified to make the tables more simple and self-explainable. The manuscript may use English editing. For example, lines 225-228: “The women who underwent instrumental birth and cesarean section were associated with feeling anxiety during the puerperium period, and those who had cesarean section stated feeling tired, like Woolhouse et al found, and sad and having postpartum depression, which agrees with Rowlands and Redshaw.” The sentence is difficult to understand. Lines 233-234: “We did not negatively associate cesarean section with urinary troubles, which disagrees with what Gundersen et al found in their study with 450,856 women in Denmark.” The sentence used triple-negative descriptions, which could be a barrier for readers.

Author Response

13 November 2019

Dear Editor of the International Journal of Environmental Research and Public Health

Thank you very much for the opportunity to revise and improve the Manuscript ID: ijerph-579499 entitled “Symptoms of discomfort and problems associated with mode of delivery during the puerperium. An observational study”.

A complete account of all points raised by the reviewers is presented in the attached report. All recommendations were taken into consideration and the necessary clarifications appear in the same order. However, if there is need to expand the manuscript to accommodate any further detail, we would be willing to do so.

We would like to thank the reviewers for their time and diligent critique, and we are looking forward to hearing from you at your earliest convenience.

Sincerely,

Juan Miguel Martinez Galiano

Reviewer (4):

Comments and Suggestions for Authors:

English language and style

(x) Extensive editing of English language and style required

Response: Dr Ingrid de Ruiter is a Medical Writer and Public Health Researcher with a background in Clinical Medicine and Public Health. Professional Member of European Medical Writers Association and American Medical Writers Association. She has carefully checked the manuscript. We believe no more grammatical errors appear.

Thank you for the invitation to review the article entitled “Symptoms of discomfort and problems associated with mode of delivery during the puerperium. An observational study” (ijerph-579499). The article aimed to determine the association between mode of delivery and the different symptoms of discomfort and maternal problems self-reported by women during the postpartum period using a cross-sectional survey study. My comments are listed below:

The authors may try to avoid multiple citations in one single sentence, especially in the section of Introduction.

Response:  Thank you for your comments. Although there are several references, all of them are related to the content of the sentence and try to support the arguments presented.

 In the method, the authors stated that “we conducted a cross-sectional study with the women attended to while giving birth in 2017 in Spain Women aged under 18 and the births that ended in fetal death antenatally were excluded.” This sentence is very confusing. Does the “Spain Women” mean Spanish women or refer to an organization or a hospital? The authors mean “…while giving birth in 2017 in Spain Women aged under 18 “, or “aged under 18 and the births that ended in fetal death antenatally were excluded”?

Response:  Thank you for your comments. We have rewritten it to clarify it “A cross-sectional study was conducted with women who gave birth in Spain in 2017 (both in public health system centers and in private centers). Births with antepartum stillbirths and women under 18 years of age were excluded.”

In the paragraph of Ethical approval, the authors should specify the hospital that granted them the IRB approval, not just using XXX.

Response: Thank you for your comments. We have corrected it “This study received approval from the Ethics Committee in Clinical Research (CEIC) of the La Mancha-Centro Hospital with ethical code 69-C.”

Table 1 consists of many Yes/No questions. For those questions, data of either “Yes” or “No” would be sufficient.

Response: Thank you for your comments We have removed the “NO” row.

The layout of Table 2 or 3 could be modified to make the tables more simple and self-explainable.

Response: Thank you for your comments. We are aware of the complexity of the tables; however, we want to provide all the information. We have not found another style to present the results.

The manuscript may use English editing. For example, lines 225-228: “The women who underwent instrumental birth and cesarean section were associated with feeling anxiety during the puerperium period, and those who had cesarean section stated feeling tired, like Woolhouse et al found, and sad and having postpartum depression, which agrees with Rowlands and Redshaw.” The sentence is difficult to understand.

Response: Thank you for your comments. We have rewritten it to clarify it “The women who had an instrumental delivery or cesarean section reported more feelings of anxiety during the puerperium period. While women who had a cesarean section stated feeling tired, in line with the results of Woolhouse et al. [41]. They also reported more feelings of sadness and having postpartum depression, which agrees with the findings of Rowlands et al. [15].”

Lines 233-234: “We did not negatively associate cesarean section with urinary troubles, which disagrees with what Gundersen et al found in their study with 450,856 women in Denmark.” The sentence used triple-negative descriptions, which could be a barrier for readers.

Response: Thank you for your comments. We have rewritten this to clarify it “We found a negative association for cesarean section and urinary problems, which is the opposite of the findings of Gundersen et al. [18] in their study with 450,856 women in Denmark. These authors identified that cesarean sections were related with a higher risk of urinary infection and the appearance of related symptoms (stinging feeling and trouble while urinating).”

Round 2

Reviewer 1 Report

Dear Authors,

Thank you for attempting to address my concerns.

I accept the manuscript in the present form.

Reviewer 2 Report

Thank you for your prompt revision; your paper is now greatly improved.

Reviewer 4 Report

The manuscript is much improved.

The format of references should be checked in accordance with the journal's requirement.